DATA RELEASE

# Draft genome of the aquatic moss *Fontinalis antipyretica* (Fontinalaceae, Bryophyta)

Jin Yu[1,2], Linzhou Li[2], Sibo Wang[2], Shanshan Dong[3], Ziqiang Chen[4], Nikisha Patel[5], Bernard Goffinet[5], Hongfeng Chen[6], Huan Liu[2] and Yang Liu[2,3,*]

1 BGI Education Center, University of Chinese Academy of Sciences, Shenzhen 518083, China
2 State Key Laboratory of Agricultural Genomics, BGI-Shenzhen & China National GeneBank, Shenzhen 518083, China
3 Laboratory of Southern Subtropical Plant Diversity, Fairy Lake Botanical Garden, Shenzhen & Chinese Academy of Sciences, Shenzhen 518004, China
4 College of Chinese Medicine Materials, Jilin Agricultural University, Changchun 130012, China
5 Department of Ecology and Evolutionary Biology, University of Connecticut, Storrs, CT 06269-3043, USA
6 South China Botanical Garden, Chinese Academy of Sciences, Guangzhou 510650, China

## ABSTRACT

Mosses comprise one of three lineages forming a sister group to extant vascular plants. Having emerged from an early split in the diversification of embryophytes, mosses may offer complementary insights into the evolution of traits following the transition to, and colonization of, land. Here, we report the draft nuclear genome of *Fontinalis antipyretica* (Fontinalaceae, Hypnales), a charismatic aquatic moss that is widespread in temperate regions of the Northern Hemisphere. We sequenced and *de novo*-assembled its genome using the 10X Genomics method. The genome comprises 385.2 Mbp, with a scaffold N50 of 45.8 Kbp. The assembly captured 87.2% of the 430 genes in the BUSCO Viridiplantae odb10 dataset. The newly generated *F. antipyretica* genome is the third moss genome, and the second seedless aquatic plant genome, to be sequenced and assembled to date.

**Subjects** Genetics and Genomics, Plant Genetics, Botany

**Submitted:** 05 May 2020

\* Corresponding author. E-mail: yang.liu0508@gmail.com

Preprint submitted at https://doi.org/10.1101/2020.04.29.069583

## DATA DESCRIPTION

With ~13,000 extant species, mosses represent perhaps the second most speciose lineage of land plants [1]. Mosses diverged from their common ancestor with liverworts [2] no later than 350 million years ago (Mya) [3–5]. The early diversification of land plants is marked by various morphological innovations, such as branching of the sporophyte or stomata [6], as well as metabolic innovations – notably biopolymers, essential materials for cuticle composition [7], which enable plants to adapt to a water-deficient, UV-exposed living environment. To date, two nuclear genomes have been sequenced for mosses; namely the model taxon and acrocarpous moss *Physcomitrium patens* [8], and *Pleurozium schreberi* [9], a representative of the diverse pleurocarpous hypnalean mosses.

*Fontinalis antipyretica* (NCBI: txid67435) is an aquatic moss species (Figure 1) from the most diverse moss order, i.e., the Hypnales [10]. Sequencing the genome of *F. antipyretica* should provide the first opportunity for a comparative genomic study in this lineage, which may have diversified after the rise of the angiosperms. Furthermore, since this is the second

**Figure 1.** Photographs of the aquatic moss *Fontinalis antipyretica*. Upper: a wild population; lower: shoots with a scale (in cm).

genome for a seedless aquatic plant, it will also allow the assessment of independent genomic transformations linked to a reversed shift to an aquatic habitat. Thus, the genome of this species would contribute to the framework necessary to study genome evolution in mosses, and to explore the adaptive transformations underlying the shifts between terrestrial and aquatic habitats.

## MATERIALS AND METHODS

A protocol collection including methods for BGISEQ-500 and 10X Genomics library construction and sequencing is available via protocols.io (Figure 2).

Fresh gametophyte tissue of *Fontinalis antipyretica* was collected in Connecticut, USA. The voucher specimen (collection number: Goffinet 14197) is deposited in the George Safford Torrey Herbarium at the University of Connecticut (CONN). Genomic DNA was extracted at the Fairy Lake Botanical Garden, and is deposited with the DNA extraction number 332.

Plant tissue was cleaned under a dissecting microscope to enhance the quality of the material. Approximately 0.4 g fresh plant shoots was ground in liquid nitrogen, and used for DNA extraction using the NucleoSpin Plant midi DNA extraction kit, following the manufacturer's protocol (Macherey-Nagel, Düren, Germany). Genomic DNA was quality-controlled using a Qubit® 3.0 Fluorometer (Thermo Fisher Scientific, USA). High molecular weight genomic DNA was used to construct 10X Genomics libraries [11] with insert sizes of 350–500 bp, following the manufacturer's protocol (Chromium Genome Chip



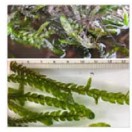

**Protocols for "Draft genome of the aquatic moss Fontinalis antipyretica (Fontinalaceae, Bryophyta)"**

Yang Liu[1], Huan Liu[1], Hongfeng Chen[2], Bernard Goffinet[3], Nikisha Patel[3], Ziqiang Chen[4], Shanshan Dong[5], Sibo Wang[1], Linzhou Li[1], Jin Yu[1]

Oct 29, 2020

[1]State Key Laboratory of Agricultural Genomics, China National GeneBank, BGI-Shenzhen, Shenzhen 518083, China, [2]South China Botanical Garden, Chinese Academy of Sciences, Guangzhou, China, [3]Department of Ecology and Evolutionary Biology, University of Connecticut, Storrs, CT, 06269-3043, USA, [4]College of Chinese Medicine Materials, Jilin Agricultural University, Changchun 130012, China, [5]Laboratory of Southern Subtropical Plant Diversity, Fairy Lake Botanical Garden, Shenzhen & Chinese Academy of Sciences, Shenzhen 518004, China

**1** Works for me    dx.doi.org/10.17504/protocols.io.bn7jmhkn

GigaScience Press    BGI

**Figure 2.** Protocol collection for the draft genome of the aquatic moss *Fontinalis antipyretica* (Fontinalaceae, Bryophyta). https://www.protocols.io/widgets/doi?uri=dx.doi.org/10.17504/protocols.io.bn7jmhkn

Kit v1, PN -120229, 10X Genomics, Pleasanton, USA) [12]. The libraries were sequenced on a BGISEQ-500 sequencer (RRID:SCR_017979) to generate 150-bp paired-end reads [13, 14].

For the genome assembly, we first calculated the distribution frequency of the barcodes in the raw data, and removed those reads containing barcodes with extremely low or high frequencies. The remaining reads were subsequently *de novo*-assembled using 10X Genomics Supernova v2.1.1 (RRID:SCR_016756) with default parameters [11]. Then, we used GapCloser v1.12-r6 (RRID:SCR_015026) to close the gaps of the preliminary assembly [15]. Default parameters were used for all software.

The genome size of *F. antipyretica* was estimated using flow cytometry. Mature leaf tissue of *Raphanus sativus*, which was cultivated from seeds obtained from the Institute of Experimental Botany (Olomouc, Czech Republic), was used for internal and external standardization. *R. sativus* has an established 2C genome size of 1.11 pg [16]. Two assays were externally standardized, and one assay was internally standardized. For each, 0.2 g of fresh tissue from the sample or the standard was used. Fresh tissue was combined with 750 µl of Cystain PI Absolute P nuclei extraction buffer (Sysmex, Kobe, Japan) in a glass petri dish, maintained on ice and chopped with a clean razor blade for 60 seconds. The internally standardized sample was co-chopped with tissue of the standard, *R. sativus*. The resulting nuclear suspension was transferred to a 30-µm CellTrics filter (Sysmex, Kobe, Japan). The flowthrough was combined with 500 µl of Cystain PI Absolute P staining solution (Sysmex, Kobe, Japan), 150 µg/mL of propidium iodide, and 50 µg/mL of RNAse. Samples were incubated on ice for 30–60 minutes. Flow cytometry was run on a BD Biosciences LSRFortessa X-20 Cell Analyzer.

Cytometry data were visualized using FlowJo v10.6.2 software (FlowJo, LLC, Ashland, OR, USA). To estimate genome size for each assay, 1C nuclei of *F. antipyretica* were compared with 2C nuclei of *Raphanus sativus*. The ratio of the mean fluorescence of the 1C *F. antipyretica* peak and the *R. sativus* 2C peak was multiplied by the genome size of *R. sativus*. The genome size estimate produced here is the mean of the estimates produced by the two externally standardized assays, as well as the one internally standardized assay.

To screen potential contamination sequences in the genome, we aligned the scaffolds against the National Center for Biotechnology Information (NCBI) nucleotide database using BLASTn with the following parameters: "-evalue 1e-5 -max_hsps 500 -num_alignments 500". In-house Perl scripts were used to assign taxonomic affiliations to

each high-scoring pair (HSP) of all query-subject pairs. Sequences identified as non-Viridiplantae origin were removed from the genome.

For genome annotation, we used Piler v1.0 (RRID:SCR_017333) [17], Repeatscout v1.0.5 (RRID:SCR_014653) [18], LTR Finder v1.0.6 (RRID:SCR_015247) [19], and RepeatMasker v4.0.6 (RRID:SCR_012954) [20] to conduct *de novo* repeat element prediction. All of the above tools were used with default parameters. RepeatMasker v4.0.6 was also implemented to identify repeats based on known repetitive sequence database, i.e., RepBase v21.01. Based on the results of repeat annotation, the genome assembly was both soft-masked and hard-masked for gene structure annotation. Gene structure annotation was performed using the MAKER v2.31.8 (RRID:SCR_005309) pipeline [21], integrating results from *ab initio* gene predictors, expressed sequence tag (EST) evidence, and protein homologs in two rounds of iterations. Augustus v3.2.1 (RRID:SCR_015981) [22], GeneMark v4.32 (RRID:SCR_011930) [23], and SNAP v2006-07-28 (RRID:005501) [24] were used for *ab initio* gene prediction. Transcriptome assembly of *F. antipyretica* was obtained from the One Thousand Plant Transcriptomes (1KP) initiative [2] and used as EST (expressed sequence tag) evidence. Protein sequences from model plant organisms and closely-related green plants, i.e., *Arabidopsis thaliana*, *Azolla filiculoides*, *Marchantia polymorpha*, *Physcomitrium patens,* and species of the *Fontinalaceae* family were selected as homolog-based evidence. Results from the first run of MAKER were used for SNAP (Semi-HMM-based Nucleic Acid Parser) training, producing a SNAP gene model, which was used by the second run of MAKER. Gene annotation results were filtered for completeness, i.e. must have complete start and stop codons by MAKER option "always_complete=1".

To reconstruct the phylogenetic tree, we used OrthoFinderv2.3.7 (RRID:SCR_017118) [25] to search for single-copy orthologs among the genomes of *F. antipyretica* and eight other green plants: *Klebsormidium nitens*, *Chara braunii*, *Anthoceros angustus*, *Marchantia polymorpha*, *Sphagnum fallax*, *Physcomitrium patens*, *Pleurozium schreberi*, and *Selaginella moellendorffii*. The genomes were downloaded from the Phytozome database [26]. A total of 472 single-copy loci were found; each locus was aligned by MAFFT v7.3.10 (RRID:SCR_011811) [27], and concatenated into one super-matrix. Finally, RAxML v8.2.4 (RRID:SCR_006086) was implemented to construct the maximum likelihood tree, using the PROTCATGTR substitution model [28]. The resulting tree was visualized using iTOL [29].

## RESULTS AND DISCUSSION

### Genome assembly and annotation

A total of 133 Gbp PE150 raw sequence data were generated by the BGISEQ-500 sequencer. The genome size of *F. antipyretica* was 385.2 Mbp, spanning 98,893 contigs, with a contig N50 of 29.7 Kbp. The final scaffold assembly included 84,391 scaffolds with an N50 length of 45.8 Kbp. Our assembly captured 87.2% of the 430 genes in the BUSCO Viridiplantae odb10 dataset [30].

The GC content of *F. antipyretica* is 40.87%, which is higher than that of *Physcomitrium patens* (i.e., 33% [8]), or *Pleurozium schreberi* (26.4% [9]). The size of the genome of *F. antipyretica* is 385.2 Mbp, which is similar to that of *P. patens* (i.e., 462.3 Mbp), but larger than that of *P. schreberi* (i.e., 318.3 Mbp). Repeats make up 51.02% of the *F. antipyretica* genome, compared with 57.0 % in *P. patens* and 28.4% in *P. schreberi*. With 16,538 genes, the gene space of the *F. antipyretica* genome is intermediary between *P. patens* with 32,926 genes and *P. schreberi* with 15,992 genes.

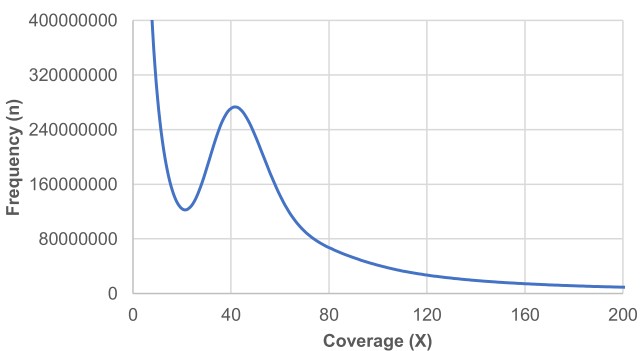

**Figure 3.** The *k*-mer distribution curve of *Fontinalis antipyretica* genome data. The curve shows a clear one-peak mode, indicating low heterozygosity and repetitive content across the genome.

## Data validation and quality control

Flow cytometry and *k*-mer analysis were used to determine the genome size of *F. antipyretica*. For flow cytometry, the nuclear peaks from which genome size was estimated comprised, on average, 242 events (see Figure 4 for a representative histogram). The mean coefficient of variance was 7.62. The mean estimated genome size is 0.484 pg. *k*-mer analysis was performed using the program Jellyfish v2.3.0 (RRID:SCR_005491) with default parameters [31]. The genome size was estimated by dividing the total *k*-mer number by the peak coverage in the *k*-mer distribution curve (Figure 3). The *k*-mer distribution curve shows one clear peak, indicating low repeat content and heterozygosity across the genome. Thus, the genome size was estimated to be 579 Mb, larger than the flow cytometry result and genome assembly. The discrepancy between genome assembly, *k*-mer estimation, and flow cytometry may be associated with contaminated next-generation sequencing (NGS) sequences used for *k*-mer calculation. Microorganism contamination may also affect the flow cytometry result.

To evaluate the completeness of the assembly, we conducted BUSCO v3.1.0 (RRID:SCR_015008) assessment on the assembly [30]. The assembly captured 87.2% complete BUSCOs of the 430 genes in the BUSCO Viridiplantae odb10 dataset.

With the streptophyte alga *K. nitens* rooted as the outgroup, bryophytes were confirmed as being a monophyletic group, and a sister group to the vascular plant *S. moellendorffii*. Consistent with previous studies [32], within bryophytes, hornwort is sister to liverworts and mosses. Within mosses, the newly sequenced *F. antipyretica* clustered as expected with another Hypnalean species, i.e., *P. schreberi* (Figure 5).

## Re-use potential

The transition of green plants from freshwater habitats to land catalyzed a major biotic diversification, which led to major climatic changes on earth. The colonization of land is characterized by the acquisition of many key innovations by plants, such as the development of an embryo, a cuticle, gravitropic detection, and pathogen defense, which were likely to be crucial for plants' survival in terrestrial environments [33]. The accumulation of genomic data, including the assembly of this moss genome, may contribute to reconstructing the evolution of the developmental networks underlying these innovations.

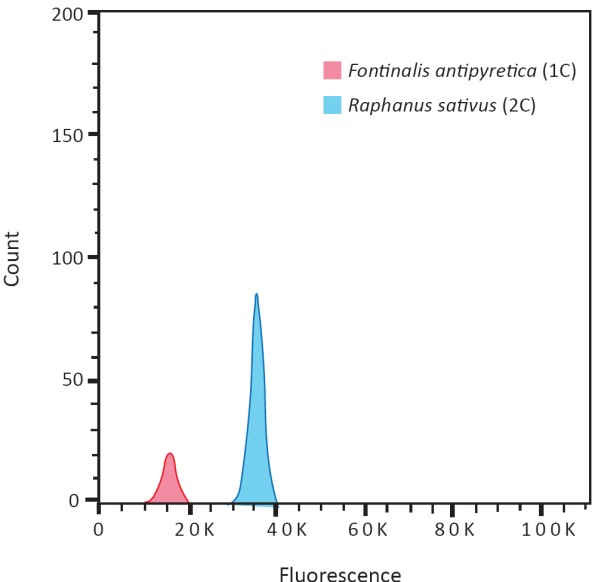

**Figure 4.** Representative sample of flow cytometry results. The 1C peak of *Fontinalis antipyretica* and the 2C peak of *Raphanus sativus* cv. Saxa are overlaid to show fluorescent intensity differences on the *x*-axis indicated by PE-A.

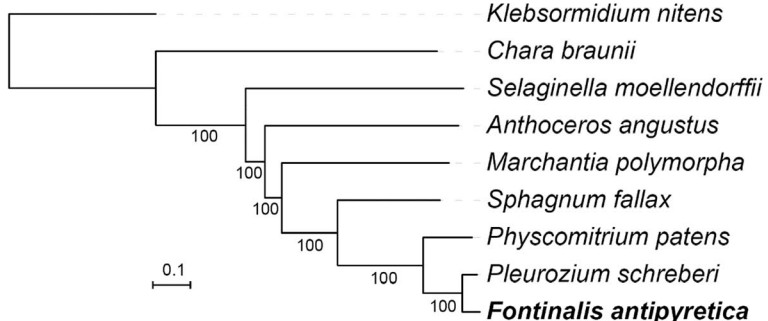

**Figure 5.** Phylogenetic tree reconstructed using nuclear genome single-copy genes, showing phylogenetic relationship of *F. antipyretica* and eight other green plants. Numbers below branches are bootstrap support values. The newly sequenced *F. antipyretica* is in bold.

Reconstructions of the relationships of extant land plant lineages are converging on a scenario in which bryophytes form a sister lineage to living vascular plants, with mosses and liverworts sharing a unique common ancestor that arose from a split from the ancestor, giving rise to hornworts [34]. Following the recent release of the hornwort genomes [32, 36], gene and gene family evolution among bryophytes can be assessed within a robust phylogenetic framework. With the resolution of the relationships between mosses [36], the accumulation of moss genomes will enable more critical estimates of trends in gene family diversity during the diversification of this lineage of land plants. Furthermore, *Fontinalis* is the first aquatic plant with a gametophyte-dominated life cycle to have its genome assembled and annotated, providing a unique opportunity to evaluate similarities in parallel adaptations in mosses, ferns [37] and angiosperms [38] following shifts to freshwater habitats.

## DATA AVAILABILITY

The raw reads have been deposited in the NCBI Sequence Read Archive (SRA; accession number PRJNA627325). The sequence reads and assemblies of the *F. antipyretica* genome have been deposited in the China National GeneBank DataBase (CNGBdb; accession number CNP0000847). Genome assemblies, protein-coding genes, and repeat annotations have been deposited in the *GigaScience* GigaDB database [39].

## DECLARATIONS
## ETHICS APPROVAL AND CONSENT TO PARTICIPATE

Not applicable.

## COMPETING INTERESTS

The authors declare that they have no competing interests.

## AUTHORS' CONTRIBUTIONS

YL, HL, and BG conceived and designed the study. BG collected the material. SD and ZC performed the experiments. YJ, ZC, LL, SW, HF and NP carried out the analyses. YJ drafted the manuscript. YJ, DS, YL, and BG revised the manuscript. All authors have read and approved the final manuscript.

## ACKNOWLEDGEMENTS

The study was funded by the Shenzhen Urban Management Bureau Fund (202005) to YL, the Strategic Priority Research Program of Chinese Academy of Sciences (XDA13020603) to HC, and the National Science Foundation (DEB-1753811) to BG.

The authors would like to thank Yang Peng and Na Li at the Shenzhen Fairy Lake Botanical Garden for laboratory assistance. This work was supported by China National GeneBank.

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
