## [Reviewer Report]

Upload additional filesDRR-20201003/form/2020.04.29.069583v1.full.pdfReviewer name and names of any other individual's who aided in reviewer Wei ZhaoDo you understand and agree to our policy of having open and named reviews, and having your review included with the published papers. (If no, please inform the editor that you cannot review this manuscript.)YesIs the language of sufficient quality?YesPlease add additional comments on language quality to clarify if needed
Are all data available and do they match the descriptions in the paper? YesAdditional CommentsAre the data and metadata consistent with relevant minimum information or reporting standards? See GigaDB checklists for examples <a href="http://gigadb.org/site/guide" target="_blank">http://gigadb.org/site/guide</a>YesAdditional CommentsIs the data acquisition clear, complete and methodologically sound?YesAdditional CommentsIs there sufficient detail in the methods and data-processing steps to allow reproduction?YesAdditional CommentsSee attached PDF fileIs there sufficient data validation and statistical analyses of data quality? NoAdditional CommentsCheck and filter potential contamination of the raw assembly.Is the validation suitable for this type of data?YesAdditional CommentsBut maybe no, see attached pdfIs there sufficient information for others to reuse this dataset or integrate it with other data?YesAdditional CommentsSee attached PDF fileAny Additional Overall Comments to the AuthorSee attached PDF fileRecommendationMajor Revision

---

## [Reviewer Report]

Reviewer name and names of any other individual's who aided in reviewer Daniel LangDo you understand and agree to our policy of having open and named reviews, and having your review included with the published papers. (If no, please inform the editor that you cannot review this manuscript.)YesIs the language of sufficient quality?YesPlease add additional comments on language quality to clarify if needed
Are all data available and do they match the descriptions in the paper? YesAdditional CommentsAre the data and metadata consistent with relevant minimum information or reporting standards? See GigaDB checklists for examples <a href="http://gigadb.org/site/guide" target="_blank">http://gigadb.org/site/guide</a>YesAdditional CommentsIs the data acquisition clear, complete and methodologically sound?YesAdditional CommentsIs there sufficient detail in the methods and data-processing steps to allow reproduction?YesAdditional CommentsIs there sufficient data validation and statistical analyses of data quality? YesAdditional CommentsThere is a exceptionally high number of scaffolds for 10x, a bad busco and a discrepancy between kmer <-> fcm&assembly size that is unusual. That would have been worthy of discussion.Is the validation suitable for this type of data?YesAdditional CommentsIs there sufficient information for others to reuse this dataset or integrate it with other data?YesAdditional CommentsAny Additional Overall Comments to the AuthorRecommendationAccept